# Out-of-Distribution Generalization in Natural Language Processing: Past, Present, and Future

**Linyi Yang**[♠♡*], **Yaoxiao Song**[♠♡*], **Xuan Ren**[♣*], **Chenyang Lyu**[△], **Yidong Wang**[♠],
**Jingming Zhuo**[♠], **Lingqiao Liu**[♣], **Jindong Wang**[♥], **Jennifer Foster**[◇], **Yue Zhang**[♠♡]

[♠] Westlake University   [♡] Westlake Institute for Advanced Study   [♣]University of Adelaide
[◇]Dublin City University   [♥] Microsoft Research Asia
[△] MBZUAI

{yanglinyi,zhangyue}@westlake.edu.cn

## Abstract

Machine learning (ML) systems in natural language processing (NLP) face significant challenges in generalizing to out-of-distribution (OOD) data, where the test distribution differs from the training data distribution. This poses important questions about the robustness of NLP models and their high accuracy, which may be artificially inflated due to their underlying sensitivity to systematic biases. Despite these challenges, there is a lack of comprehensive surveys on the generalization challenge from an OOD perspective in natural language understanding. Therefore, this paper aims to fill this gap by presenting the first comprehensive review of recent progress, methods, and evaluations on this topic. We further discuss the challenges involved and potential future research directions. By providing convenient access to existing work, we hope this survey will encourage future research in this area.

## 1 Introduction

Pre-trained Language Models (PLMs) (Devlin et al., 2018; Liu et al., 2019b; Radford et al., 2018) have revolutionized natural language processing (NLP) and enabled remarkable advances in Large-scale Language Models (LLMs) (Touvron et al., 2023; Gozalo-Brizuela and Garrido-Merchan, 2023; Pichai, 2023) . Despite substantial progress in developing accurate models in several natural language understanding tasks, including sentiment analysis (Kaushik et al., 2019; Ni et al., 2019; Yang et al., 2021; Lu et al., 2022; Luo et al., 2022a,b), natural language inference (Williams et al., 2018), and machine reading comprehension (Kaushik and Lipton, 2018; Sugawara et al., 2020), a major challenge persists – out-of-distribution (OOD) generalization – which entails the ability of a model to accurately classify text instances from distributions different from those of the training data (Ben-David et al., 2010; Hendrycks and Gimpel, 2017; Hupkes et al., 2022). This paper aims to provide a comprehensive overview of the current state of research in OOD generalization for natural language understanding, highlighting key methodologies, advancements, and unique challenges.

The importance of OOD generalization in NLP cannot be overstated, as real-world data often exhibit diversity and unpredictability. Numerous applications, such as sentiment analysis, document categorization, and spam detection (Shen et al., 2021; Yang et al., 2022), necessitate models capable of adapting to novel and unforeseen data distributions. While machine learning models generally demonstrate strong in-distribution performance, their performance frequently deteriorates when confronted with OOD instances, underscoring the need for effective strategies that facilitate generalization beyond the training distribution.

Although research on OOD generalization in NLP is emerging, it is not on the scale of other tasks like computer vision (Ye et al., 2021; Koh et al., 2021) and time series (Du et al., 2021b; Gagnon-Audet et al., 2022). Furthermore, most related surveys in NLP focus on measuring and improving model robustness against adversarial attacks (Schlegel et al., 2020; Arora et al., 2021), or providing causal explanations (Keith et al., 2020). Among them, Wang et al. (2021d) is the most relevant review to this paper, but their work does not differentiate between data-level variance and shortcut features and also not discuss LLMs.

To address these limitations, this survey provides an extensive examination of the existing literature on OOD generalization in NLP, covering a diverse array of techniques and approaches. We focus on two perspectives of OOD generalization: the data distribution, which is model-independent and the feature distribution, which is model-oriented. Ad-

---

[1]These authors contributed equally to this work.

[2]"Large Language Models (LLMs)" refers to recent generative models while "Pre-trained Language Models refers to small-scale pre-trained models" in this paper.

ditionally, we discuss the evaluation metrics and benchmarks employed to assess the effectiveness of these techniques, as well as the limitations and drawbacks of current methodologies.

Throughout this survey, we trace the evolution of OOD generalization techniques in natural language processing, from the early approaches based on traditional machine learning algorithms to more recent advancements driven by deep learning architectures, also including the discussion of the most recent emergent abilities of LLMs. We identify the key innovations and breakthroughs that have shaped the field, while also highlighting areas where progress has been relatively slow or incremental. Our analysis emphasizes the interconnected nature of these advancements and the importance of driving fundamental research in the generalization problem towards unforeseen data distributions. In addition, this survey aims to identify open challenges and future directions for OOD generalization in NLP, especially for LLMs. We discuss the limitations of current techniques, potential avenues for improving model robustness and adaptability, and emerging research trends that may contribute to the development of more effective OOD generalization strategies.

The remainder of this survey is organized as follows: we formalize the scope of OOD generalization in Section 2. Then, we propose a novel taxonomy towards OOD robustness and review existing methodologies developed for addressing OOD issues in Section 3. In particular, we identify two salient aspects of OOD generalization, namely *Data Variance* and *Shortcut Features*. We outline two representative application scenarios in Section 4, namely *Deployment in High-stake Domains* and *Social Bias*. We also introduce the methods for improving the OOD robustness in Section 5 before discussing the redefinition of OOD in the era of large language models.

## 2 The Scope of OOD Generalization

Denote a set of labeled data as $\mathcal{D} = \{(x_i, y_i)\}_{i=1}^{N}$, where an input $x \in X$, output $y \in Y$, and $N$ is the number of datasets. A training dataset $\mathcal{D}_{train} = \{(X_{train}, Y_{train})\}$ is generated by sampling from $\mathcal{D}$ with distribution $\mathcal{P}_{train}$, and the test dataset $\mathcal{D}_{test} = \{(X_{test}, Y_{test})\}$ is sampled from $\mathcal{D}$ with distribution $\mathcal{P}_{test}$. **Out-of-distribution (OOD)** refers to the circumstance when $\mathcal{P}_{train} \neq \mathcal{P}_{test}$.

In the context of text classification, let $\mathcal{X}$ be the set of all possible documents, $\mathcal{Y}$ be the set of all possible labels, and $D$ be a training distribution defined on $\mathcal{X} \times \mathcal{Y}$. Suppose the true target distribution is $P_{\mathcal{X},\mathcal{Y}}$, which is close to but not identical to $D$ with $P_{\mathcal{X},\mathcal{Y}} \neq D$. When we encounter a document that is drawn from a distribution $Q_{\mathcal{X}}$ that is significantly different from $P_{\mathcal{X}}$, we refer to it as an out-of-distribution (OOD) sample. An OOD sample may have a vocabulary or language not presenting in $P_{\mathcal{X}}$.

A text classification model $f : \mathcal{X} \to \mathcal{Y}$ is considered OOD if its performance on $Q_{\mathcal{X}}$ is significantly worse than on $P_{\mathcal{X}}$ due to the distribution shift. The OOD detection function can be derived from a probabilistic perspective using Bayesian inference. In this case, we can estimate the posterior probability of a document being OOD given its bag-of-words features through Bayesian model averaging:

$$
\begin{aligned}
P(\text{OOD}|\boldsymbol{x}) &= \sum_{\theta} P(\text{OOD}|\theta, \boldsymbol{x}) P(\theta|\boldsymbol{x}) \\
&= \sum_{\theta} \frac{P(\boldsymbol{x}|\text{OOD}, \theta) P(\text{OOD}|\theta) P(\theta)}{P(\boldsymbol{x})}
\end{aligned}
$$

where $\theta$ denotes the model parameters, $\boldsymbol{x}$ is the bag-of-words representation of a document, $P(\text{OOD}|\theta)$ is the prior probability of the model being OOD assuming the model parameter $\theta$, $P(\boldsymbol{x}|\text{OOD}, \theta)$ is the likelihood of observing the bag-of-words features $\boldsymbol{x}$ given that the document is OOD and the model parameter $\theta$, $P(\theta)$ is the prior probability of the model parameter $\theta$, and $P(\boldsymbol{x})$ is the marginal likelihood of observing the bag-of-words features $\boldsymbol{x}$.

The conditional OOD probability of model $f_{\theta}$ on input $\boldsymbol{x}$ given the parameter $\theta$ is defined as:

$$
P(\text{OOD}|\theta, \boldsymbol{x}) = \frac{P(\boldsymbol{x}|\text{OOD}, \theta) P(\text{OOD}|\theta)}{P(\boldsymbol{x}|\theta)}.
$$

As can be seen from the above equations, OOD can be perceived in terms of both the data- and model- levels, robust models can be more resistant to data variances. The OOD detection function can be defined as a threshold on the posterior OOD probability:

$$
g(\boldsymbol{x}) = [\max_{y} P(y|\boldsymbol{x}) < \epsilon],
$$

where $P(y|\boldsymbol{x})$ is the posterior probability of the document belonging to class $y$ given the bag-of-words features $\boldsymbol{x}$, and $\epsilon$ is a threshold parameter that determines the confidence of the prediction.

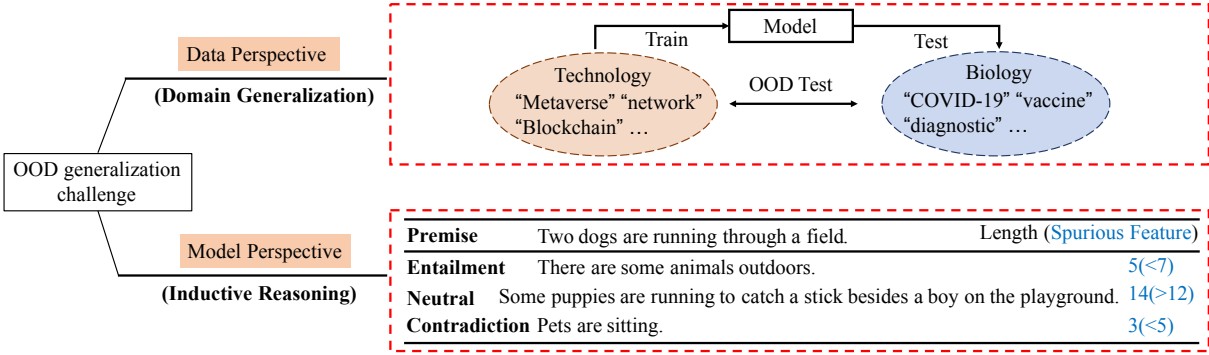

Figure 1: Taxonomy of OOD generalization scope and examples.

The OOD detection performance can be evaluated using metrics such as the Receiver Operating Characteristic (ROC) curve or the Kolmogorov-Smirnov (KS) statistic, which capture the trade-off between true positive rate and false positive rate, or the maximum distance between the cumulative distribution functions of the OOD and in-distribution predictions, respectively.

## 3 Taxonomy of Out-of-Distribution Problems

We classify OOD problems into two perspectives, as depicted in Figure 1, namely *Data* and *Features*. Data variance encompasses the domain generalization problem, while "shortcut features" represent a range of issues typically caused by shortcut learning, which cannot be avoided from inductive reasoning.

### 3.1 Data

Data variance can be seen as a typical problem of domain generalization methods, assuming the unavailability of labeled or unlabeled data from the target domain. Previous studies have explored this approach in sentiment analysis (SA) (Kaushik et al., 2019; Ni et al., 2019; Yang et al., 2021; Lu et al., 2022), natural language inference (NLI) (Williams et al., 2018; Hendrycks et al., 2020), and named entity recognition (NER) (Jia and Zhang, 2020; Plank, 2021). Different domains have intrinsically different feature distributions, and instances from different domains have different predicted vocabulary distributions, which leads to the OOD generalization challenge, as shown in Figure 1.

Numerous NLP studies aim to tackle systematic variations between training and testing distributions, encompassing a vast body of literature on domain generalization (Blitzer et al., 2006; Ganin et al., 2016; Ruder and Plank, 2018; Han and Eisenstein, 2019; Guo et al., 2020) and cross-task transfer (Johnson et al., 2017; Levy et al., 2017; Eriguchi et al., 2018; Wang et al., 2022). These studies can be broadly categorized into input-level variation and output-level variation. Notable comprehensive surveys dedicated to this task include those by Ramponi and Plank (2020) and Wang et al. (2021d) but fail to decouple data and features.

**Compositional generalization** refers to the challenge of learning the distribution of atoms given the surface distributions of their compositions. It has garnered significant attention in NLP research, encompassing areas such as semantic parsing (Iyer et al., 2017; Gupta et al., 2022), QA (Gu et al., 2021; Lewis et al., 2021), machine translation (Li et al., 2021), and general natural language understanding (NLU) tasks (Lake and Baroni, 2018; Keysers et al., 2020). Researchers (Keysers et al., 2020; Kim et al., 2021) have found that state-of-the-art neural models struggle to generalize to novel compounds in a manner similar to human performance. Several benchmarks have been introduced to evaluate compositional generalization. For example, the SCAN dataset by Lake and Baroni (2018) is designed for sequence-to-sequence generalization (Russin et al., 2019; Li et al., 2019a; Gordon et al., 2019; Andreas, 2020). Additionally, Keysers et al. (2020) and Kim and Linzen (2020) propose the CFQ and COGS benchmarks, respectively, for semantic parsing. Li et al. (2021) propose the CoGnition dataset to assess how neural machine translation models generalize to novel compounds (Hupkes et al., 2020; Zheng and Lapata, 2021; Dankers et al., 2021; Jung, 2022).

To address the challenges of compositional generalization, achieving OOD robustness is highly desirable as current NLP models have shown **fragility** to variations in expression, where even minor punctuation changes can lead to different outputs (Wang et al., 2021c). Furthermore, Moradi et al. (2021) observe significant performance decay of NLP mod-

els in domain-specific tasks, such as the clinical domain, due to noise, grammar errors, missing words, punctuation, typos, and other factors. Additionally, Wang et al. (2021c) develop a unified multilingual robustness evaluation platform for NLP tasks to provide comprehensive robustness analysis.

Another source of OOD data is human-crafted **adversarial data**. For example, the recently proposed contrast sets (Kaushik et al., 2019; Gardner et al., 2020; Warstadt et al., 2020) reveal the failure of capturing true underlying distributions, which show the fragility of models against small variations of input expressions. In addition, although researchers also propose a benchmark to reveal the importance of OOD detection (Hendrycks and Gimpel, 2017; Hendrycks et al., 2020; Fort et al., 2021), there is a consensus that we still lack a standard definition of OOD examples and fine-grained evaluations. A full survey of current available OOD datasets can be found in Appendix A.

### 3.2 Features

Models' predictions are often influenced by shortcut features learned from spurious patterns between training data and labels, as well as existing shortcuts in the dataset. For instance, as illustrated in Figure 1 (Model Perspective), sentence length has inadvertently become a learned feature during training, where 60% of the hypotheses in entailment examples have 7 or fewer tokens and half of the hypotheses with more than 12 or fewer than 5 tokens are neutral or contradiction, respectively (Gururangan et al., 2018).

Ideally, a model should learn rational (Jiang et al., 2021; Lu et al., 2022) features for robust generalization. Take sentiment classification for example. In order to decide a positive polarity for the sentence "I like this movie.", a rationale feature should be "like" rather than "movie". The latter is referred to as a spurious feature (Kaushik et al., 2020), which leads to reduced generalization. Other cases of feature issues include shortcut features (Geirhos et al., 2020). For instance, in machine reading comprehension, if the question asks for a date and the input passage contains only one date, a model can bypass a reasoning process and directly use the date feature for output (Lai et al., 2021). For numerical (Hendrycks et al., 2020; Wang et al., 2021a; Cobbe et al., 2021) and logical (Yu et al., 2019b; Liu et al., 2021c) reasoning tasks, the rationale feature should be the underlying

algebraic and logic deduction, which turn out to be extremely challenging to learn using existing pre-trained models, leading to weak generalization.

Current NLP methods tend to learn implicitly superficial cues instead of the causal associations between the input and labels, as evidenced by (Geirhos et al., 2020; Guo et al., 2023b), and thus usually show their brittleness when deployed in real-world scenarios. Recent work (Sugawara et al., 2018, 2020; Lai et al., 2021; Wang et al., 2021b; Du et al., 2021a; Zhu et al., 2021; Bastings et al., 2021) indicates that current PLMs unintentionally learn shortcuts to trick specific benchmarks and such tricks (i.e., syntactic heuristics, lexical overlap, and relevant words) that use partial evidence to produce unreliable output, which is particularly serious in the open domain.

## 4 Application Scenarios

We highlight the importance of OOD generalization in two real-world application scenarios, in which low OOD robustness may lead to serious consequences.

### 4.1 Deployment in Practical Domains

Despite the generalization ability of LLMs, such as ChatGPT (OpenAI, 2023b), the relatively low generalization ability of medium-size models hinders the deployment of NLP systems, especially for high-stake domains, from health and medicine to finance and business (Imbens and Rubin, 2015; Choi et al., 2023), and should be taken more seriously. Notably, a recent comprehensive evaluation of OOD generalization in text classification named GLUE-X (Yang et al., 2022) shows that the average accuracy of PLMs on cross-domain evaluations falls significantly short of human performance, even for the highest-performing model (**80.1% – human versus 74.6% – model**). In contrast to GLUE, where over 20 single-model results outperform human baselines, none of the baselines, including InstructGPT and ChatGPT, considered in GLUE-X is able to surpass human performance using OOD tests. The lack of sufficient OOD generalization ability is also related to social bias.

### 4.2 Social Bias

Recent studies (Gardner et al., 2020) have uncovered a problematic tendency for gender bias in sentiment analysis (Zmigrod et al., 2019; Maudslay et al., 2019; Lu et al., 2020). Bias exists

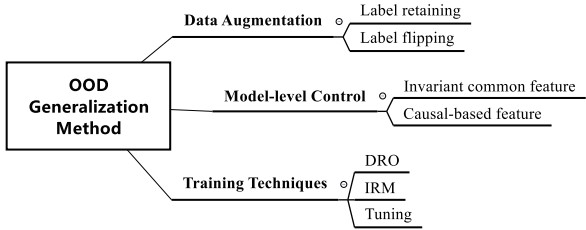

Figure 2: Classifying methods regarding the OOD generalization problem.

in different forms of language representations, including word embeddings (Bolukbasi et al., 2016; Caliskan et al., 2017; Zhao et al., 2018b; Gonen and Goldberg, 2019), contextualized word embeddings (Zhao et al., 2019) and sentence embeddings (May et al., 2019). Some found that the embeddings of feminine words and masculine words are often clustered into different groups (e.g., occupation) (Bolukbasi et al., 2016; Caliskan et al., 2017; Zhao et al., 2018b, 2019; Gonen and Goldberg, 2019). Gender bias also affects coreference resolution systems, which tend to link a pronoun to occupations dominated by the pronoun gender (Rudinger et al., 2018; Zhao et al., 2018a). In machine translation, Vanmassenhove et al. (2018) and Stanovsky et al. (2019) find that models tend to make stereotypical assignments of gender roles when translating occupation words. Apart from gender bias, there are other forms of social bias in NLP data, such as disability (Hutchinson et al., 2020), race (Kiritchenko and Mohammad, 2018), age (Diaz et al., 2018), etc.

## 5 Methods

Existing work to address OOD issues in NLP can be categorized into three groups: data augmentation (Sec.5.1), model-level control (Sec.5.2), and training approaches (Sec.5.3) shown in Fig. 2. Descriptions of current OOD generalization methods categorized by tasks are introduced in the Appendix.

### 5.1 Data Augmentation

Data augmentation (DA) techniques are employed to enhance the diversity of training data without the need for explicitly collecting new data (Feng et al., 2021). This approach proves beneficial for improving the generalization of NLP models by reducing overfitting and enhancing robustness. Several existing surveys have discussed data augmentation in low-resource NLP scenarios from different perspectives (Hedderich et al., 2021; Feng et al., 2021;

Bayer et al., 2021; Chen et al., 2021a; Li et al., 2022). In this study, our focus is on data augmentation regarding OOD generalization.

**Semi-fact Data Augmentation.** One common type of data augmentation method in NLP involves substituting part of the content or introducing perturbations to the original data, primarily focusing on enhancing the diversity without altering the semantic meaning or label. Synonym substitution has been explored by Zhang et al. (2015), Miao et al. (2020) and Yue and Zhou (2020) to replace words or entities. Perturbation techniques typically involve manipulating tokens within sentences (Zhang et al., 2018; Wei and Zou, 2019; Miao et al., 2020; Xie et al., 2020; Zhao et al., 2019, 2018a), as well as adversarial perturbations (Miyato et al., 2017; Cheng et al., 2019; Zhu et al., 2019; Jiang et al., 2020; Zheng et al., 2020), which employ large pre-trained models (e.g., GPT-2, BART, BERT) for generating conditional data augmentations. Lu et al. (2022) apply the human-in-the-loop technique incorporated with semi-fact data augmentation for improving the OOD robustness of PLMs in sentiment analysis.

**Counterfactual data augmentation** (CDA) is widely adopted to mitigate bias in neural NLP tasks by operating on biased text (Maudslay et al., 2019; Zmigrod et al., 2019). A counterfactual example constructed by flipping the label helps to learn real associations between input and label. For instance, Lu et al. (2020) proposes a CDA method to mitigate gender bias in neural coreference resolution, which is a generic methodology for corpus augmentation via causal interventions (i.e., breaking associations between gendered and gender-neutral words). In text classification, Kaushik et al. (2019), Kaushik et al. (2020), and Wang and Culotta (2020) employ humans for generating counterfactual data, which has been shown to be effective to mitigate the influence of spurious patterns. Automatic counterfactual generation aims to change the data distribution of the training data so that models can alleviate reliance on dataset-specific bias and exclude spurious correlations (Yang et al., 2021; Wang and Culotta, 2021; Wu et al., 2021) and has been improved in a recent work (Fan et al., 2023) by using data-level and sentence-level constraints.

### 5.2 Model-level Operations

Feature representation learning holds a pivotal role in OOD generalization. In this section, we evaluate model-level approaches, focusing on two critical

aspects: invariance and the causal factor.

**Invariant Common Features** Research on invariant features as a means to facilitate transfer learning has been an enduring pursuit in the field. In the context of discrete linear models, various methods have been developed to harness data from the target domain to aid representation learning. For instance, Structured Correspondence Learning utilizes unlabeled target-domain data to establish mappings between features across different domains (Blitzer et al., 2006). On a similar note, Daumé III (2009) employs labeled data for this purpose.

Additionally, Johnson and Zhang (2005) also uses unlabeled data, but in a different setting. Transitioning to neural models, adversarial learning emerges as a prevalent technique (Goodfellow et al., 2015; Ganin et al., 2016; Zhang et al., 2019a). In this approach, an adversarial loss function is employed to train a domain classifier. This classifier attempts to eliminate domain-specific information in the hidden layers, thereby producing representations that are more amenable for cross-domain (Liu et al., 2018; Li et al., 2019b; Du et al., 2020) or cross-task decision making (Johnson et al., 2017; Levy et al., 2017; Eriguchi et al., 2018; Lee et al., 2019; Wang et al.; Keung et al., 2019; Vernikos et al., 2020; Wang et al., 2022). In sentiment analysis, Liu et al. (2018) and Du et al. (2020) conduct adversarial training to derive enhanced domain-invariant features for cross-domain classification.

Feature clustering and other techniques are also adopted to learn invariant features, which requires OOD generalization on unseen tasks. For instance, Johnson et al. (2017), Arivazhagan et al. (2019), Ji et al. (2020), Liu et al. (2021a) train translation models for better learning of language-independent representations, which help the model generalize to unseen language pairs. More recently, Yin et al. (2022) categorize source contextualized representations to boost compositional generalization.

**Causal-based Features** Causal inference aims to determine the effectiveness of one variable on another variable (Holland, 1986; Morgan and Winship, 2015; Imbens and Rubin, 2015; Pearl et al., 2000). Because the relationships between the causal features and the labels are invariant under distribution shift (Pearl et al., 2000; Quionero-Candela et al., 2009), learning causal relationships allows a model to acquire robust knowledge that holds beyond the distribution of a set of training tasks or the observed data (Schölkopf et al., 2021).

In addition, learning a causal model requires fewer examples to adapt to new environments (Schölkopf et al., 2021).

There has been much research on using causal inference to improve OOD generalization. For instance, in social media, Pryzant et al. (2018) induce a lexicon that is helpful for target label prediction yet uncorrelated to a set of confounding variables, and Saha et al. (2019) perform propensity score-based causal analysis on social media posts for evaluating the effect of psychiatric medications.

### 5.3 Training Approaches

In the presence of distribution shifts, optimization tends to be influenced by irrelevant signals, resulting in severe failures when applied to OOD test data (Liu et al., 2021b). Consequently, there has been significant interest in recent work regarding training techniques.

**Distributionally Robust Optimization (DRO)** aims to learn a model on the worst-case distribution scenario (domain) while expected to generalize well on test data. To improve the worst-case domain, Sagawa et al. (2020) propose a group DRO method that requires explicit group annotation of samples. Methods based on group DRO and its variants have recently been applied in NLP tasks, such as NLI (Sagawa et al., 2020; Liu et al., 2021b), machine translation (Zhou et al., 2021a), spoken language understanding (Broscheit et al.), and toxicity detection (Michel et al., 2020). For example, Oren et al. (2019) design a DRO procedure for generative modeling that minimizes the simulated worst-case distribution scenario over the mixture of topics. Zhou et al. (2021c) consider the worst-case with language pairs to optimize multilingual neural machine translation.

**Invariance Risk Minimization (IRM)** Different from DRO, which focuses on domain shift robustness, IRM methods focus on learning invariant representations. IRM (Arjovsky et al., 2019) is a recently proposed learning paradigm that estimates non-linear, invariant, causal predictors from multiple training environments for improving OOD generalization. It has several advantages. For example, it does not need extra knowledge to manipulate the original data (e.g., human intervention or rule-based methods) and extra large computation. Existing work has studied the IRM and its variants in NLP. Choe et al. (2020) investigate IRM on synthetic settings and simple MLP and machine

learning models in sentiment analysis. Dranker et al. (2021) study OOD generalization for NLI by IRM, in which environments are constructed by ensuring whether the dataset and bias are synthetic or naturalistic. Peyrard et al. (2021) propose a language framework based on IRM-games (Ahuja et al., 2020) for learning invariant representations that generalize better across multiple environments. The OOD objective in learning the causal invariance can also be viewed as a multi-objective optimization problem, which has been explored by Chen et al. (2023b) using a pareto learning strategy.

**Tuning** Three popular tuning approaches for preserving the pre-trained features are reviewed: *prompt tuning*, *adapter tuning*, and *linear probing*.

*Adapter tuning* (Rebuffi et al., 2017; Houlsby et al., 2019) contains a few task-specific trainable parameters and are injected between layers of frozen pre-trained models. Training only the adapter modules can help models achieve competitive performance on various tasks, such as multi-task text classification (Houlsby et al., 2019), NER (Pfeiffer et al., 2020), multi-task QA (Friedman et al., 2021), and multilingual speech translation (Le et al., 2021).

*Prompt tuning* (Liu et al., 2021f) methods convert the downstream problems into language modeling problems. It adds prompt tokens as the prefix to the questions and converts them to input texts, then use a pre-trained language model to process the input texts in order to generate the answer sequences. There are two variations of prompt tokens, hard prompt tokens, and soft prompt tokens. Tuning hard prompt tokens requires fine-tuning the pre-trained models (Petroni et al., 2019; Cui et al., 2021). Tuning soft prompt tokens only need to fine-tune the prompt tokens, thus preserving the pre-trained features (Li and Liang, 2021; Lester et al., 2021; Qin and Eisner, 2021; Liu et al., 2021g). Soft prompt tuning is helpful for a wide range of cross-domain tasks, such as NER (Chen et al., 2021c, 2022b), text classification (Gao et al., 2021; Zhong et al., 2021a; Utama et al., 2021), table-to-text (Li and Liang, 2021), QA and paraphrase detection (Lester et al., 2021) and NLU (Liu et al., 2021g).

*Linear probing* (Liu et al., 2019a) fine-tunes the top layers while keeping the lower layers frozen. Compared to full fine-tuning, linear probing performs better for OOD generalization but reaches lower accuracy on IID data. Kumar et al. (2022) propose a two-step strategy, which first trains the model with linear probing and then performs fine-tuning (LP-FT). This approach has been theoretically proven to improve both in-domain and OOD performance for deep neural models.

# 6 Large Language Models

Large language models (LLMs) have attracted increasing attention in the field of artificial intelligence recently. However, as a crucial property towards artificial general intelligence (AGI), the OOD robustness is still under-explored (Wang et al., 2023b). Given its importance, we review the recent work on the OOD generalization of LLMs.

**OOD Definition** It is of imminent importance to reframe the OOD definition in the era of LLM dominance since the pre-trained corpora of LLMs are not publicly available. The absence of pre-trained corpus information makes it hard to define OOD examples for LLMs in NLP. Although providing an accurate and strict definition remains challenging for large foundation models, researchers make attempts to build label-sharing OOD data for LLMs from two perspectives, namely, *synthetic data*, and *distribution shift over time*. Synthetic data is generally defined as artificially annotated information generated by algorithms or simulations, which can be hand-crafted as challenging OOD examples for LLMs. Distribution shift over time refers to the idea of using real-world datasets collected after 2021 as OOD test data, which is the latest data collection time of ChatGPT (Wang et al., 2023b).

Another type of OOD data refers to the task of generalizing to unseen classes. For instance, in open-set label shift (Garg et al., 2022), the test data includes examples from novel classes not present in the training data, making it impossible for classical small models to predict correctly. LLMs such as ChatGPT can alleviate this issue by using in-context learning, as evidenced by recent research (Xu et al., 2022). This means that LLMs can be used to improve robustness with minimal human intervention but they cannot fully solve this problem and open-set label shift remains challenging.

**OOD Detection** Previous research on OOD detection has employed models to identify test examples that come from a different distribution (Hendrycks and Gimpel, 2017; Hendrycks et al., 2018). Some of these approaches introduce new training objectives, such as using a contrastive objective (Winkens et al., 2020; Zhou et al., 2021c). When the type of distribution shift is known, the

model can be trained to exhibit uncertainty when presented with known OOD examples (Hendrycks et al., 2020). However, the distribution of SOTA LLMs, such as ChatGPT and GPT-4 is hidden and cannot be inferred. Very recently, CoNAL (Xu et al., 2022) provides an alternative for generating novel examples which simulate open-set shifts and has proven to be effective for OOD detection.

Regarding language models (LLMs), the deepfake detectors aimed at distinguishing content generated by humans or LLMs is closely related to previous algorithms designed for OOD detection (Guo et al., 2023a). When it comes to deepfake detection, one intuitive approach is to employ statistical boundaries that differentiate linguistic patterns between human-written and machine-generated text (Mitchell et al., 2023). However, these statistical methods have a limitation: they assume access to the model's prediction distributions is possible, which hinders their application to models behind APIs, such as ChatGPT. An alternative paradigm involves training neural-based detectors (Bakhtin et al., 2019; Fagni et al., 2021), including the official implementation of OpenAI (OpenAI, 2023a).

**OOD Robustness** Previous studies have extensively examined various aspects of ChatGPT in the domains of law (Choi et al., 2023), ethics (Shen et al., 2023), reasoning (Bang et al., 2023) and planning (Yao et al., 2023). However, limited attention has been given to its robustness (Kawaguchi et al., 2017) against out-of-distribution (OOD) inputs. Evaluating OOD robustness in a reliable manner poses a significant challenge due to the massive and unknown training data of LLMs. Wang et al. (2023b) offers an initial investigation into the robustness of ChatGPT by presenting OOD results on Flipkart and DDXPlus. Building on this work, Ye et al. (2023) delimit the robustness of LLMs in comparison to conventional models, with a focus on aligning the threat model to the realistic deployment of LLMs. Additionally, Zhu et al. (2023) measures LLMs' resilience to adversarial prompts using adversarial textual attacks on character, word, sentence, and semantic levels. Collectively, these evaluations raise similar concerns regarding the limited robustness of ChatGPT. Among different attack levels, character-level attacks demonstrate higher robustness, while word-level attacks pose the greatest vulnerability. Recently, we have noticed that several papers survey the robustness issue in Large Language Models (LLMs) from differ-

ent perspectives, including factuality (Wang et al., 2023a), hallucination (Rawte et al., 2023), and evaluation methods (Chang et al., 2023).

## 7 Future Directions and Conclusion

We consider multiple promising directions for improving the OOD robustness from four perspectives: (1) enhancing the learning of such salient **causal features**, either by the help of human guidance (Kaushik et al., 2019; Lu et al., 2022) via human-in-the-loop or through psychologically inspired neural structures (Chowdhery et al., 2022), can be worthy of consideration; (2) **data-centric AI:** both the selection of training data and the careful design of prompt learning have proven effective in domain generalization (Chen et al., 2022c). In addition, the emerging ability of large-scale language models holds a huge potential for OOD generalization, benefiting from the instruction tuning, which requires a high-quality data construction process; (3) **alignment methods:** this can be effectuated through the deployment of reinforcement learning algorithms, be it in an online or offline setting (Christiano et al., 2017; Chen et al., 2023a); (4) **neuro-symbolic modeling for NLP:** purely neural models like ChatGPT can possess incredibly powerful generalization abilities. While it is more-or-less accepted that purely neural models face challenges of reasoning beyond surface-level patterns. In order to avoid picking up spurious correlations, neuro-symbolic approaches are proposed to improve the models' OOD robustness by combining the learning capabilities of neural networks with the expressive power of symbolic reasoning (Alon et al., 2022; Jung et al., 2022; Manhaeve et al., 2018; Hamilton et al., 2022).

This paper presents an ambitious attempt to categorize the challenges of OOD generalization, focusing on both data and model levels. By undertaking this categorization, our aim is to shed light on the limitations of current methods, emphasize the crucial nature of OOD robustness, and provide quick access to existing references for further exploration. In addition, we emphasize the ongoing significance of OOD robustness in the era of large language models, emphasizing the need to address this aspect. We call upon researchers in the NLP community to delve deeper into the proper definition of OOD in the context of large models and develop appropriate benchmark tests that accurately measure the OOD generalization ability of LLMs.

## 8 Limitations

When we categorise OOD-related NLP work, we mostly focus on the recently appearing papers, which can be retrospected to classical generalization studies. Moreover, the literature on domain generalization and domain adaptation has not been distinguished in this work. Lastly, the introduction of classical transfer learning algorithms has not been included for the time being.

## Acknowledgement

This publication has emanated from research conducted with the financial support of the Pioneer and "Leading Goose" R&D Program of Zhejiang under Grant Number 2022SDXHDX0003 and the 72nd round of the Chinese Post-doctoral Science Foundation project 2022M722836. Yue Zhang is the corresponding author.

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

# A Appendix

In an effort to clearly outline the various challenges tied to OOD generalization, we divide our discussion into two distinct aspects, represented by Table 1 (Data Distribution) and Table 2 (Feature Distribution), respectively. In Table 1, we focus on issues that arise due to differences and changes in data. It systematically lists the ways in which variability in data attributes can make it difficult for models to effectively generalize to out-of-distribution samples in different tasks. In general, we categorize the reference materials from two perspectives: annotation artifacts (label-sharing generalization) and output variance (label-different generalization). Different from label-sharing generalization approaches, which rely on few-shot or unlabeled data from the target domain, label-different generalization is based on zero-shot learning using clustering and other techniques. We also outline the typical datasets for each task and corresponding representative methods.

On the other hand, we concentrate on a different set of issues from the feature perspective. These problems originate from the models' tendencies to learn from spurious patterns or "shortcut features" in the data, which might not reflect the true underlying relationships between inputs and labels, leading to the generalization challenge. Ideally, a model should learn rational features. However, inductive reasoning naturally relies on patterns and trends from the training data. This reliance can result in models performing well on familiar data but poorly when faced with new, OOD examples. The OOD generalization challenge can not be avoided when using deep learning based approaches, yet it can be alleviated by several techniques as illustrated in Table 2. Collectively, Table 1 and Table 2 provide a thorough understanding of the challenges in OOD generalization, and set the stage for developing strategies to address these issues.

To provide a fine-grained description of OOD generalization methods in NLP, we introduce key points of representative methods in different tasks from Tables 3-5, ranging from the scope and method to dataset and metric. We hope these materials can serve as a quick access to existing references for further exploration.

| Data Variance | Task | Papers | Key Methods | Typical Datasets |
|---|---|---|---|---|
| Annotation artifacts | Sentiment Analysis | Ganin et al. (2016); Chen and Cardie (2018); Laparra et al. (2020). | Adversarial learning ; Multinomial adversarial networks. | Amazon Reviews; IMDB Reviews. |
| | MT | Belinkov and Bisk (2017); Khayrallah and Koehn (2018). | Invariant representation learning; Training on adversarial examples. | WiCoPaCo; RWSE Wikipedia; Revision Dataset; MERLIN corpus. |
| | Label-sharing NER | Liu et al. (2021d); Huang et al. (2021a). | Noisy supervised pre-training; Calibrated confidence methods. | CoNLL2003; Tweet; Webpage; Wikigold. |
| | QA | Cai et al. (2017); Min et al. (2019); Bartolo et al. (2021b,a); Lyu et al. (2022). | Generator-in-the-loop models. | ROC Story; HOTPOTQA; NewsQA; SQuAD1.1; AdversarialQA. |
| | NLI | Poliak et al. (2018); Naik et al. (2018); Zellers et al. (2018); Feng et al. (2019); McCoy et al. (2019); Le Bras et al. (2020); Sakaguchi et al. (2020); Nie et al. (2020); Liu et al. (2020); Gardner et al. (2021); Pezeshkpour et al. (2022); Wu et al. (2022). | Data augmentation; Human-and-model-in-the-loop; Adversarial filtering; Training on adversarial examples. | Stress Test; ANLI; SWAG; HANS; SNLI-hard; MultiNLI-hard. |
| | MRC | Kaushik and Lipton (2018); Sugawara et al. (2018, 2020); Bartolo et al. (2020); Lai et al. (2021). | Shortcut investigation. | bAbI; SQuAD; CBT; CNN; Whodid-What; DuoRc. |
| | MT | Vanmassenhove et al. (2018); Stanovsky et al. (2019); Tomalin et al. (2021); Choubey et al. (2021). | Adversarial learning; Gender-filtered self-training. | WinoMT; MuST-SHE. |
| | Coreference Resolution | Rudinger et al. (2018); Zhao et al. (2018a). | Data augmentation. | WinoBias; Winogender Schemas. |
| | Toxicity Detection | Park et al. (2018); Dixon et al. (2018). | Data augmentation; Debias word embeddings. | Sexist Tweets (st); Abusive Tweets (abt). |
| Output variance | Label-Different NER | Snell et al. (2017); Ghaddar and Langlais (2017); Wu et al. (2020); Nguyen et al. (2021); Cui et al. (2021); Ma et al. (2021); Zhou et al. (2021b); Lee et al. (2021); Das et al. (2021); Wang et al. (2022). | Self-training methods; Prompt-based methods; Information theories; Prototype-based methods; Distance-based methods; Knowledge-enhanced methods. | CoNLL2003; MIT Movie; MIT Restaurant; WNUT2017; Ontonotes 5.0 Dataset; BioNER. |
| | Machine Translation | Johnson et al. (2017); Zhang et al. (2020); Arivazhagan et al. (2019); Ji et al. (2020); Liu et al. (2021a). | Multilingual corpus pre-training; Back translation; Invariance representation learning; Language independent representations learning. | WMT'14; WMT'17; Newstest 2012; Newstest 2013; Newstest 2016; Newstest 2015; IWSLT 2017. |

Table 1: OOD generalization challenges related to the data variance.

| Flexibility of Expression | Task | Papers | Key Methods | Typical Datasets |
|---|---|---|---|---|
| Compositional generalization | Text Classification | Oren et al. (2019); Hendrycks et al. (2020); Wang et al. (2021b); Du et al. (2021a); Liu et al. (2021b); Moradi and Samwald (2021); Náplava et al. (2021); Wang et al. (2021c). | Data augmentation; Regularization on shortcuts; Spurious features identification & removal; Distributionally robust optimization (DRO). | WildNLP; TextFlint; IMDB Reviews; Kindle Reviews. |
| | Natural Language Generation | Cheng et al. (2019); Zhang et al. (2019b); Zhou et al. (2021a); Hewitt et al. (2021). | Adversarial attack learning; Group DRO; Robust fine-tuning. | NIST; WMT'14; WevNLG; XSUM; Open-domain QA. |
| | Evaluations | Czarnowska et al. (2019); Kaushik et al. (2019); Gardner et al. (2020); Warstadt et al. (2020); Hu et al. (2020); Lewis et al. (2020); Lazaridou et al. (2021); Liu et al. (2021e); Koh et al. (2021); Chen et al. (2022a). | Contrast sets; Fine-grained evaluations. | BLiMP; XTREME; MLQA; ARXIV; Wilds; SQuAD. |
| | NLU | Lake and Baroni (2018); Russin et al. (2019); Li et al. (2019a); Gordon et al. (2019); Andreas (2020); Keysers et al. (2020); Kim and Linzen (2020); Kim et al. (2021). | Dedicated train objects; Structure annotation. | SCAN; CFQ; COGS. |
| | Semantic Parsing | Iyer et al. (2017); Lake and Baroni (2018); Dong and Lapata (2018); Lake (2019); Yu et al. (2019a); Furrer et al. (2020); Kim (2021); Gupta et al. (2022). | Span-level supervised attention; Human-in-the-loop; Meta sequence-to-sequence learning; Structurally diverse sampling. | ATIS; GEO; SCAN; CFQ. |
| | Machine Translation | Chen et al. (2020); Li et al. (2021); Zheng and Lapata (2021). | Neural symbolic stack machines; Representation disentanglement. | CoGnition; SCAN. |
| | QA | Gu et al. (2021); Lewis et al. (2021); Bogin et al. (2021). | Data augmentation; Prompt-tuning; Continual pre-training. | GRAILQA; TriviaQA; Open Natural Questions; WebQuestions. |
| Logic reasoning | MRC | Dong and Lapata (2016); Yu et al. (2019b); Rogers et al. (2021); Liu et al. (2021c); Zhong et al. (2021b); Huang et al. (2021b). | GAN; Graph neural networks; Knowledge-enhanced methods. | SQuAD; DROP; LogiQA; HotpotQA; ReClor; AR-LAST. |
| | Mathematical Problem | Brown et al. (2020); Cobbe et al. (2021); Drori et al. (2021); Hendrycks et al. (2021) | Self-supervised training (GPT3); Training verifiers; Program synthesis (Codex). | MATH Datasets; DeepMind Datasets. |

Table 2: OOD generalization challenges related to shortcut features learned by models.

| Work | Task | Scope | Method | Dataset | Metric |
|------|------|-------|--------|---------|--------|
| Dong and Lapata (2016) | MRC | Logical Reasoning (Domain Variance) | Propose an attention-enhanced encoder-decoder model invariant representation | JOBS, GEO, ATIS, IFTTT | Accuracy |
| Yu et al. (2019b) | MRC | Logical Reasoning (Bias) | Introduce a new Reading Comprehension dataset requiring logical reasoning (ReClor) extracted from standardized graduate admission examinations. | ReClor | Accuracy |
| Liu et al. (2021c) | MRC | Logical Reasoning (Bias) | Introduce a comprehensive dataset which is sourced from expert-written questions. | Logiqa | Accuracy |
| Zhong et al. (2021b) | MRC | Logical Reasoning (Bias) | Introduce a new dataset consisting of questions from the Law School Admission Test from 1991 to 2016. | AR-LSAT | Accuracy |
| Kaushik and Lipton (2018) | MRC | Annotation artifacts | Establish sensible baselines for the bAbI, SQuAD, CBT, CNN, and Who-did-What datasets, finding that question- and passage-only models often perform surprisingly well. | bAbI, SQuAD, CBT, CNN, Whodid-What | Accuracy |
| Sugawara et al. (2018) | MRC | Annotation artifacts | Establish sensible baselines for the bAbI, SQuAD, CBT, CNN, and Whodid-What datasets, finding that question- and passage-only models often perform surprisingly well. | QA4MRE, CNN/Daily Mail, Children's Book, WikiReading, LAMBADA, Who-did-What, ProPara, CliCR, SQuAD, DuoRC | Accuracy |
| Sugawara et al. (2020) | MRC | Annotation artifacts (shortcut) | Propose a semi-automated, ablation-based methodology to evaluate capacity of MRC datasets. | CoQA,DuoRC, HotpotQA, SquAD, SQuAD, ARC, MCTest, MultiRC, RACE, SWAG | Accuracy F1 |
| Bartolo et al. (2020) | MRC | Annotation artifacts (shortcut) | Propose an adversarial annotation data collection method. Training on adversarially collected samplesleads to strong generalization. | SQuAD1.1 | F1 |
| Lai et al. (2021) | MRC | Annotation artifacts (shortcut) | Propose two synthetic dataset and two new method to investigate shortcut in MRC especially on paraphrasing. | QWM-Para dataset derived from SQuAD | F1 |
| Cheng et al. (2019) | NLG | Data noise | Propose double adversarial input MT model to improve the robustness. | LDC corpus, NIST, WMT'14, newstest2013,2014 | BLEU score |
| Zhang et al. (2019b) | NLG | Annotation artifacts (exposure bias) | In word-level sampling context words is not only from the ground truth sequence but also from the predicted sequence by the model during training, where the predicted sequence is selected with a sentence-level optimum. | NIST, WMT'14 | BLEU score |
| Zhou et al. (2021a) | NLG | Annotation artifacts (domain) | Propose a new learning objective for MNMT based on DRO. | 58-languages TED talk corpus, WMT | BLEU score |
| Hewitt et al. (2021) | NLG | Annotation artifacts (domain) | Present methods to combine the benefits of full and lightweight finetuning, achieving strong performance both ID and OOD. | WebNLG, XSUM, Open-domain QA | BLEU score ROUGE-2 score Exact match accuracy |

Table 3: Methods towards OOD generalization challenge in the task of MRC and NLG.

| Work | Task | Scope | Method | Dataset | Metric |
|---|---|---|---|---|---|
| Jia et al. (2019) | NER | Input variance | Design cross-domain and cross- task network for NER domain generalization. | CoNLL, BioNLP13PC, BioNKP13CG, CBS Newws | F1 |
| Jia and Zhang (2020) | NER | Input variance | Multi-task learning with multi-cell LSTM for NER domain generalization. | CoNLL2003, Broad Twitter, Twitter, BioNLP13PC, BioNLP13CG, CBS News | F1 |
| Liu et al. (2021h) | NER | Input variance | Introduce a cross-domain NER dataset with a domain-related corpus and propose a baseline. | CoNLL2003, CrossNER | F1 |
| Chen et al. (2021b) | NER | Input variance | Data Augmentation for crossdomain NER. Propose a novel neural architecture to transform the data representation from a high-resource to a low-resource domain. | Ontonotes 5.0, Temporal Twitter | F1 |
| Ghaddar and Langlais (2017) | NER | Output variance | Propose a large, high quality, annotated corpus WiNER for cross-domain NER. | CoNLL, MUC, ONTO, WGOLD, WEB | F1 |
| Vu et al. (2020) | NER | Output variance | Adversarially trained masked LMs with domain generalization. | CoNLL2003, WNUT2016, FIN, JNLPBA, BC2GM, BioNLP09, BioNLP11EPI | F1 |
| Wu et al. (2020) | NER | Output variance | Propose a teacher-student learning method fro cross-linguial NER. | CoNLL-2002, CoNLL-2003 | F1 |
| Nguyen et al. (2021) | NER | Output variance | Cross domain zero shot NER with knowledge base. | music, science datset | F1 |
| Cui et al. (2021) | NER | Output variance | Propose a template-based method for NER, treating NER as a language model ranking problem in a sequence-to-sequence framework, where original sentences and statement templates filled by candidate named entity span are regarded as the source sequence and the target sequence. | CoNLL, MIT Movie Review, MIT Restaurant Review | F1 |
| Ma et al. (2021) | NER | Output variance | Reformulate NER tasks as LM problems without templates. | CoNLL2003, Ontonotes 5.0, MIT-Movie | F1 |
| Zhou et al. (2021b) | NER | Output variance | Propose Masked Entity Language Modeling (MELM) as a novel data augmentation framework for low-resource NER to alleviate the token-label misalignment. | CoNLL | F1 |
| Lee et al. (2021) | NER | Output variance | Propose a demonstration-based learning method for NER, which lets the input be prefaced by task demonstrations for in-context learning. | CoNLL-2003, Ontonotes 5.0, BC5CDR | F1 |
| Das et al. (2021) | NER | Output variance | Propose a novel contrastive learning technique that optimizes the inter-token distribution distance instead of class-specific attributes for Few-Shot NER. | OntoNotes, CoNLL'03, WNUT '17, GUM | F1 |
| Wang et al. (2022) | NER | Output variance | Propose an information theoretic perspective method to imporve out-of-vocabulary entities prediction. | WNUT2017,TwitterNER,BioNER, Conll03-Typos, Conll03-OOV | F1 |
| Liu et al. (2021d) | NER | Data noise | Propose a calibrated confidence estimation and integrate it into a self-training framework for boosting performance in general noisy settiings. | CoNLL, Tweet, Webpage, Wikigold | F1 |
| Gu et al. (2021) | QA | Compositional generlization (Bias) | Construct new large-scale, high-quality dataset GrailQA, and propose a novel BERT-based KBQA model. | GRAILQA | F1 |
| Lewis et al. (2021) | QA | Compositional generlization (Bias) | Evaluate three popular open-domain benchmark datasets and find that all models perform dramatically worse on questions that cannot be memorized from training sets. | WebQuestions, TriviaQA, Open Natural Questions | Exact match score |
| Bogin et al. (2021) | QA | Compositional generlization (Bias) | Propose a model that computes a representation and denotation for all question spans in a bottom-up, compositional manner using a CKY-style parser. Inductive bias towards tree structures dramatically improves systematic generalization to out-of-distribution examples. | arithmetic expressions benchmark, CLEVR, CLOSURE | F1 |
| Cai et al. (2017) | QA | Compositional generalization | Propose a hierarchical RNN with attention to encode the sentence in the story and score candidate endings. | ROC Story | Accuracy |
| Min et al. (2019) | QA | Compositional generalization | Propose a single-hop BERT-based RC model. | HOTPOTQA | F1 |
| Bartolo et al. (2021b) | QA | Compositional generalization (domain) | Introduce a generator-in-the-loop model to provide real-time suggestions for annotator, which maintains the advantages of DADC and reduce annotation cost. | SQuAD1.1, AdversarialQA, GAA-assisted data | Median time per example, Validated Model Error Rate (vMER), Median time per validated model-fooling example, Downstream effectiveness (F1 score) |
| Lyu et al. (2022) | QA | Compositional generalization (domain) | Extend the scope of "OOD" by splitting QA examples into different subdomains according to their several internal characteristics including question type, text length, answer position. Examine the performance of QA systems trained on the data from different subdomains. | SQuAD 1.1, NewsQA | F1 |

Table 4: Methods towards OOD generalization challenge in the task of NER and QA.

| Work | Task | Scope | Method | Dataset | Metric |
|---|---|---|---|---|---|
| Wang et al. (2021b) | SA | Annotation artifacts (shortcut) | Automatically identify such spurious correlations in NLP models at scale. | SST, Yelp, Occupation dataset, Amazon Kitchen, Amazon Electronics | Precision Importance score |
| Kaushik et al. (2019) | SA, NLI | Input variance | CDA. | SNLI, IMDB | Accuracy |
| Kaushik et al. (2020) | SA, NLI | Input variance | evaluate the efficacy of CDA. | IMBb, Yelp, Amazon, Semeval, CRD, SNLI, MultiNLI | Accuracy |
| Hendrycks et al. (2020) | SA, NLI | Input variance | evaluate OOD generalization of pre-trained model. | SST-2,Yelp Review,Amazon Review,MultiNLI | Accuracy |
| Wang and Culotta (2020) | SA | Input variance | Train spurious feature detector & improve OOD generalization. | IMDB reviews, Kindle reviews, Toxic comment,Toxic tweet | Accuracy |
| Wang and Culotta (2021) | SA | Input variance | train spurious feature detector & Improve robustness to spurious correlations via CDA. | IMDB reviews, Amazon, Kindle reviews | Accuracy |
| Yang et al. (2021) | SA | Input variance | CDA & improve OOD genralization. | SST-2, IMDB, Amazon Reviews, Semeval 2017, Yelp Reviews | Accuracy |
| Lu et al. (2022) | SA | Input variance | improving robustness via auto Semi-factual data augmentation | IMDb, Amazon reviews, Yelp reviews, SST, SemEval-2017 Twitter. | Accuracy |
| Chen and Cardie (2018) | SA | Data noise | improving OOD generalization via learning invariant features. | Amazon reviews, FDU-MTL dataset | Accuracy |
| Johnson et al. (2017) | MT | Output variance | zero-shot MT via training on multilingual corpus | WMT'14, WMT'15. | BLEU score |
| Zhang et al. (2020) | MT | Output variance | improve zero-shot MT: enforce translation to the target language via backtranslation. | OPUS-100 | BLEU score Win ratio |
| Arivazhagan et al. (2019) | MT | Output variance | improve zero-shot MT: learn invariant representations via auxiliary losses. | newstest-2012, WMT14, newstest-2013, IWSLT 2017 | BLEU score |
| Ji et al. (2020) | MT | Output variance | improve zero-shot MT: obtain an universal encoder for different languages. | Europarl, MultiUN | BLEU score |
| Liu et al. (2021a) | MT | Output variance | improve zero-shot MT: removing residual connections. | IWSLT 2017, Europarl v7, PMIndia | BLEU score |
| Zheng and Lapata (2021) | MT | Compositional generalization | Improve composional generalization: propose an extension to sequence-to-sequence models which encourages disentanglement. | COGS, CFQ | BLEU score Exact match score Compound translation error rate |
| Belinkov and Bisk (2017) | MT | Data noise | Increase model robustness: structure-invariant word representations & robust training. | IWSLT 2016, WiCoPaCo, Wikipedia Revision Dataset, The MERLIN corpus, Czech: manually annotated essays | BLEU score |
| Stanovsky et al. (2019) | MT | Annotation artifacts | present the challenge set for evaluating gender bias in machine tranlation. | WinoMT | Accuracy F1 |
| Choubey et al. (2021) | MT | Annotation artifacts | propose gender-filtered self-training (GFST) to improve gender translation accuracy. | WinoMT, MuST-SHE | Accuracy F1 Recall BLEU score |
| Williams et al. (2018) | NLI | Input variance | introduce MultiNLI benchmark. | MultiNLI, SNLI | Accuracy |
| Naik et al. (2018) | NLI | Annotation artifacts | propose Stress Test dataset for NLI. | MultiNLI | Accuracy Error rate |
| Zellers et al. (2018) | NLI | Annotation artifacts | propose dataset SWAG for measuring common reasoning of NLI model. | SWAG, SNLI | Accuracy |
| Feng et al. (2019) | NLI | Annotation artifacts | We illustrate how partial-input baselines can overshadow trivial. | SNLI | Accuracy |
| McCoy et al. (2019) | NLI | Annotation artifacts | Introduced HANS dataset which contains three fallible syntactic heuristics. | MultiNLI, HANS | Accuracy |
| Le Bras et al. (2020) | NLI | Annotation artifacts | Use AFLITE to reduce dataset biases, thus improve OOD generalization. | SNLI, ANLI, HANS, NLI-Diagnostics, Stress tests, QNLI, MultiNLI | Accuracy |
| Sakaguchi et al. (2020) | NLI | Annotation artifacts | Introduce WINOGRANDE, which is harder & larger than Winograd Schema Challenge. | WINOGRANDE, WSC, DPR , COPA, KnowRef, Winogender | Accuracy |
| Nie et al. (2020) | NLI | Annotation artifacts | Introduce ANLI, collected via iterative&adversarial human-and-model-in-the-loop procedure. | ANLI, SNLI, MultiNLI, SNLI-Hard, NLI Stress Tests | Accuracy Error rate |
| Liu et al. (2020) | NLI | Annotation artifacts | derive adversarial examples in terms of the hypothesis-only bias and explore eligible ways to mitigate such bias. | SNLI, MultiNLI | Accuracy |
| Wu et al. (2022) | NLI | Annotation artifacts | generating debiased datasets through filter out instances contribute to spurious correlations. | SNLI, MultiNLI, HANS, SNLI-hard, MultiNLI-hard | Accuracy |
| Du et al. (2021a) | NLI | Annotation artifacts (shortcut) | Propose a shortcut mitigation framework LTGR using knowledge distillation framework, to suppress the model from making overconfident predictions for samples with large shortcut degree. | MultiNLI, FEVER, and MultiNLI-backdoor | Accuracy |
| Liu et al. (2021b) | NLI | Annotation artifacts (domain) | Propose a simple two-stage approach, that minimizes the loss over a reweighted dataset (second stage) where we upweight training examples that are misclassified at the end of a few steps of standard training (first stage). It overcome the requirement of expensive group annotations in group DRO. | MultiNLI, CivilComments-WILDS | Accuracy |
| Oren et al. (2019) | Text classification | Annotation artifacts (bias) | Propose a new DRO based approach called topic conditional value at risk. | Yelp, ONEBWORD, TPIPADV | perplexity |

Table 5: Methods towards OOD generalization challenge in the task of SA, NLI, and MT.