# OpenReview forum: "Out-of-Distribution Generalization in Natural Language Processing: Past, Present, and Future"
_EMNLP/2023/Conference — EMNLP 2023 Main_

### Official Review · Reviewer_Z5rz · 2023-08-04

**Soundness:** 3

**Excitement:**

3: Ambivalent: It has merits (e.g., it reports state-of-the-art results, the idea is nice), but there are key weaknesses (e.g., it describes incremental work), and it can significantly benefit from another round of revision. However, I won't object to accepting it if my co-reviewers champion it.

**Paper Topic And Main Contributions:**

The main contributions of the paper are:
- The authors provide a comprehensive overview of the problem of out-of-distribution generalization in NLP, discussing its history, current state, and future prospects.
- They shed light on the significant challenges faced by ML systems in NLP when dealing with OOD data.
- The paper stimulates further discussion about the robustness of NLP models and the need for more accurate measures of their performance.

**Reasons To Accept:**

The strengths of the paper include:
- The paper addresses a crucial problem in NLP, making it highly relevant to current research in the field.
- The authors provide a comprehensive overview of the issue, considering its past, present, and future, which can be very informative for other researchers in the field.
- The paper stimulates further discussion and research into the robustness of NLP models and the challenges of dealing with OOD data.

**Reasons To Reject:**

- It's unclear whether the authors have proposed any concrete solutions or strategies to address the problem of out-of-distribution generalization in NLP.
- The paper might be too focused on the problem itself, without providing sufficient practical insights or guidelines for dealing with it

**Reproducibility:**

2: Would be hard pressed to reproduce the results. The contribution depends on data that are simply not available outside the author's institution or consortium; not enough details are provided.

**Reviewer Confidence:**

3: Pretty sure, but there's a chance I missed something. Although I have a good feel for this area in general, I did not carefully check the paper's details, e.g., the math, experimental design, or novelty.

---

> ### Author Rebuttal · Authors · 2023-08-28
>
> Thank you for your constructive suggestions.
>
> Question A: It's unclear whether the authors have proposed any concrete solutions or strategies to address the problem of out-of-distribution generalization in NLP.
>
> Answer: Our team has a rich history of working on OOD generalization in NLP, with over six methodology papers to our credit. However, due to EMNLP's anonymity policy, we are unable to list specific papers in this rebuttal. Our in-depth analysis of these issues inspired us to compile this survey.
>
> Question B: The paper might be too focused on the problem itself, without providing sufficient practical insights or guidelines for dealing with it.
>
> Answer: As highlighted in Section 5 (Methods), we have outlined three primary strategies to enhance the OOD generalization capabilities of models. Furthermore, we plan to include additional quantitative results related to these methods in the updated version of our paper.

---

### Official Review · Reviewer_t4je · 2023-08-05

**Typos Grammar Style And Presentation Improvements:** N/A
**Soundness:** 4

**Excitement:**

4: Strong: This paper deepens the understanding of some phenomenon or lowers the barriers to an existing research direction.

**Missing References:**

N/A

**Paper Topic And Main Contributions:**

This paper provides a comprehensive review of research on out-of-distribution (OOD) generalization in natural language processing. The key contributions are:
1. It proposes a novel taxonomy to categorize OOD issues into data perspective (domain generalization) and feature perspective (shortcut learning).
2. It comprehensively summarizes and compares different methods for improving OOD generalization, including data augmentation, model-level operations, and training approaches.
3. It highlights the importance of OOD robustness in real-world applications such as high-stake domains and for mitigating social biases.
4. It discusses redefining OOD in the era of large language models and reviews recent work evaluating their OOD capabilities.

**Questions For The Authors:**

- A. When presenting the various methods, it would be useful to have some empirical comparisons of their effectiveness on benchmark OOD tasks.
- B. The paper mentions limitations of current OOD datasets and evaluations. What are some ways the community could improve OOD benchmarks and metrics to better assess model generalization?

**Reasons To Accept:**

- The taxonomy of OOD issues from data and model view is insightful and provides a structured way to understand this problem.
- The paper comprehensively covers a wide range of tasks, datasets, and evaluation metrics related to OOD generalization.

**Reasons To Reject:**

- More analysis comparing the effectiveness of different methods on benchmark datasets would be useful.
- The limitations of current OOD datasets and evaluation practices could be discussed further.

**Reproducibility:**

N/A: Doesn't apply, since the paper does not include empirical results.

**Reviewer Confidence:**

3: Pretty sure, but there's a chance I missed something. Although I have a good feel for this area in general, I did not carefully check the paper's details, e.g., the math, experimental design, or novelty.

---

> ### Author Rebuttal · Authors · 2023-08-28
>
> We appreciate the thoughtful review and suggestions.
>
> Question A: When presenting the various methods, it would be useful to have some empirical comparisons of their effectiveness on benchmark OOD tasks.
>
> Answer: Thank you for the suggestion. In the next version of the paper, we will provide empirical comparisons of the mentioned methods using the WILDS and GLUE-X benchmarks to demonstrate their effectiveness.
>
> Question B. The paper mentions the limitations of current OOD datasets and evaluations. What are some ways the community could improve OOD benchmarks and metrics to better assess model generalization?
>
> Answer: We thank the reviewer for pointing out this pivotal question for future research. Indeed, developing advanced OOD benchmarks and metrics is crucial for the future of both OOD and language model research.
>
> Based on the experience from this survey paper, we believe there should be at least two aspects for future research:
>
> (1) Diversifying Metrics: Current OOD benchmarks only focus on one metric (often accuracy). As language models grow in size and face increasingly complex tasks, it becomes infeasible to rely solely on one metric to gauge OOD performance. For instance, the holistic evaluation work [1] offers a good example of how to design diverse and challenging benchmarks for LM evaluation, which could extend to the OOD area.
>
> (2) Designing Anti-causal Benchmarks: Anti-causal benchmarks would involve reversing the cause-effect relationship, where the effect is used to predict the cause. Introducing anti-causal benchmarks could challenge existing systems with the most difficult examples. A model that can perform well on both causal and anti-causal benchmarks is likely to be more robust and less sensitive to changes in data distribution. Such a feature is particularly crucial for OOD generalization.
>
> We will elaborate further on this topic in the updated version of the paper.
>
> [1] Liang, P., Bommasani, R., Lee, T., Tsipras, D., Soylu, D., Yasunaga, M., Zhang, Y., Narayanan, D., Wu, Y., Kumar, A. and Newman, B., 2022. Holistic evaluation of language models. arXiv preprint arXiv:2211.09110.

---

### Official Review · Reviewer_DYuH · 2023-08-09

**Soundness:** 4

**Excitement:**

4: Strong: This paper deepens the understanding of some phenomenon or lowers the barriers to an existing research direction.

**Paper Topic And Main Contributions:**

This paper is a survey on out-of-distribution (OOD) generalization. It provides a definition and describes many works in the fields of domain adaptation with a focus on more recent papers, while addressing multiple approaches in a structured way. Two main contributions of the present survey is the distinction between the data and model perspectives and the discussion of recent large language models in the context of OOD generalization.

**Reasons To Accept:**

- The survey paper is well-written, structured and comprehensive.
- It addresses a very important topic and challenge in natural language processing.

**Reasons To Reject:**

- The concepts of "Pretrained Language Models"and "Large-scale Language models" are not clearly defined. It is not clear whether the latter only refers to recent generative models. A third expression, "large language models", is also used. Also, the place of Large Language models in the structure of the paper is confusing: Section 6 is dedicated to Large Language Models but their use is already described in Section 4.1. I think Section 6 should also include more details about the current knowledge, as reported by the cited papers.

- It would be useful to further connect between previous research in domain adaptation and the current challenges faced by large language models (it is done a little bit in lines 653-665).

**Reproducibility:**

N/A: Doesn't apply, since the paper does not include empirical results.

**Reviewer Confidence:**

3: Pretty sure, but there's a chance I missed something. Although I have a good feel for this area in general, I did not carefully check the paper's details, e.g., the math, experimental design, or novelty.

**Typos Grammar Style And Presentation Improvements:**

Presentation:
- The different concepts used in the context of pretrained language models (e.g. pretraining, finetuning, prompting) should be defined and explained in more details.
- I think Figure 1 should be explained in more details in the text, in particular for the Data Perspective.
- line 217: "fail to decouple" - Maybe "do not decouple" is more suitable here.
- lines 292-297: it would be interesting to discuss the possible consequence of the shortcut described in reading comprehension in the case there are multiple answer candidates and/or no answer (See for example the notions of "plausible answers"/ "competitive entities" , respectively in "Know What You Don’t Know: Unanswerable Questions for SQuAD. Pranav Rajpurkar, Robin Jia and Percy Liang, Proc. of ACL 2018" and in "Zero-shot Event Extraction via Transfer Learning: Challenges and Insights. Qing Lyu, Hongming Zhang, Elior Sulem and Dan Roth, Proc. of ACL 2021").
- line 624: I think that the findings of Wang et al., 2023 should be presented. This is particularly relevant to Section 6.

Typo:
- line 71: not discuss -> do not discuss
- lines 323-324: the sentence is incomplete.

---

> ### Author Rebuttal · Authors · 2023-08-28
>
> Thanks for the constructive comment.
>
> Question A: The concepts of "Pre-trained Language Models" and "Large-scale Language models" are not clearly defined.
>
> Answer: We agree with your point that the concepts of "Pre-trained Language Models" and "Large-scale Language models" need to be further distinguished. To address this issue, we will use a unified name, ‘Large Language Models (LLMs)’, referring to recent generative models.
>
> Question B: The place of Large Language Models in the structure of the paper is confusing: Section 6 is dedicated to Large Language Models but their use is already described in Section 4.1.
>
> Answer: We will merge the description of Section 4.1 into Section 6. We will also provide a more detailed overview of the current progress of LLMs in Section 6.
>
> Question C: It would be useful to further connect previous research in domain adaptation and the current challenges faced by large language models (it is done a little bit in lines 653-665).
>
> Answer: Based on the taxonomy presented in [1], previous domain adaptation algorithms are classified into three categories: data manipulation, representation learning, and learning strategies. For closed models like ChatGPT or GPT-4 in the era of LLMs, representation learning and learning strategies cannot be easily manipulated. However, it is still worth exploring whether previous methods can be effectively applied to open LLMs like LLaMA.
>
> [1] Wang, J., Lan, C., Liu, C., Ouyang, Y., Qin, T., Lu, W., Chen, Y., Zeng, W. and Yu, P., 2022. Generalizing to unseen domains: A survey on domain generalization. IEEE Transactions on Knowledge and Data Engineering.
>
>
> Question D: lines 292-297: it would be interesting to discuss the possible consequence of the shortcut described in reading comprehension in the case there are multiple answer candidates and/or no answer [2, 3].
>
> Answer: Thanks for the reminder! We will add the citation of SQuAD 2.0 and zero-shot event extraction accordingly.
>
> [2] Know What You Don’t Know: Unanswerable Questions for SQuAD. Pranav Rajpurkar, Robin Jia and Percy Liang, Proc. of ACL 2018
>
> [3] Zero-shot Event Extraction via Transfer Learning: Challenges and Insights. Qing Lyu, Hongming Zhang, Elior Sulem and Dan Roth, Proc. of ACL 2021
>
>
> Question E: line 624: I think that the findings of Wang et al., 2023 should be presented. This is particularly relevant to Section 6.
>
> Answer: Thanks for your suggestions. In the work of Wang et al., 2023 [4], the authors found that the ID-OOD performance does not always correlate positively with ChatGPT and text-davinci-003. They also demonstrated that there's significant room for improvement in ChatGPT, especially from an adversarial and out-of-distribution perspective. However, it's important to note that this preliminary evaluation relies on a small sample size, so the conclusions should be approached with caution.
>
> [4] Wang, J., Xixu, Hu.,  Hou, W., Chen, H., Zheng, R., Wang, Y., Yang, L., Ye, W., Huang, H., Geng, X. and Jiao, B., 2023, April. On the Robustness of ChatGPT: An Adversarial and Out-of-distribution Perspective. In ICLR 2023 Workshop on Trustworthy and Reliable Large-Scale Machine Learning Models.
>
> Typos:
> Thank you for pointing out these issues. We will revise our manuscript accordingly.

---

### Meta-Review · Area_Chair_VqfZ · 2023-09-07

**Recommendation:** 4

**Metareview:**

The paper presents a survey on the problem of out-of-distribution generalization in natural language processing. The authors propose a novel taxonomy to categorize OOD challenges, describe multiple approaches in a structured way and also discuss recent large language models in the context of OOD generalization. In particular, the paper stimulates further discussion about the robustness of NLP models and the need for more accurate measures of their performance. Thus, it could be very helpful and impactful for researchers working on that topic.

Pros / Strengths:
- Survey addresses a crucial problem in NLP
- Well-written und well-structured
- Novel taxonomy is helpful in understanding the problem
- A wide range of tasks, datasets, and evaluation metrics related to OOD generalization are covered in the paper
- Stimulates further discussion about the robustness of NLP models and how to measure their performance

Cons / Weaknesses:
- Some concepts are not defined clearly enough
- There is no quantitiative analysis of methods on benchmark datasets

Action items for improving the paper:
- Concepts around recent LLMs need to be defined more clearly
- Limitations of current OOD datasets and evaluation practices should be discussed in more detail
- Practical insights and recommended ways of dealing with different OOD issues should be better highlighted throughout the paper

---

### Decision · Program_Chairs · 2023-10-07

**Decision:**

Accept-Main

**Comment:**

The paper presents a survey on the problem of out-of-distribution generalization in natural language processing. The authors propose a novel taxonomy to categorize OOD challenges, describe multiple approaches in a structured way and also discuss recent large language models in the context of OOD generalization. In particular, the paper stimulates further discussion about the robustness of NLP models and the need for more accurate measures of their performance. Thus, it could be very helpful and impactful for researchers working on that topic.

Pros / Strengths:
- Survey addresses a crucial problem in NLP
- Well-written und well-structured
- Novel taxonomy is helpful in understanding the problem
- A wide range of tasks, datasets, and evaluation metrics related to OOD generalization are covered in the paper
- Stimulates further discussion about the robustness of NLP models and how to measure their performance

Cons / Weaknesses:
- Some concepts are not defined clearly enough
- There is no quantitiative analysis of methods on benchmark datasets

Action items for improving the paper:
- Concepts around recent LLMs need to be defined more clearly
- Limitations of current OOD datasets and evaluation practices should be discussed in more detail
- Practical insights and recommended ways of dealing with different OOD issues should be better highlighted throughout the paper